# A Customized Extended Kalman Filter for Removing the Impact of the Magnetometer’s Measurements on Inclination Determination

**DOI:** 10.3390/s23249756

**Published:** 2023-12-11

**Authors:** Yang Chen, Hailong Rong

**Affiliations:** 1School of Microelectronics and Control Engineering, Changzhou University, Changzhou 213164, China; chenyangcczu@cczu.edu.cn; 2School of Mechanical Engineering and Rail Transit, Changzhou University, Changzhou 213164, China

**Keywords:** extended Kalman filter, magnetic and inertial measurement units, inclination determination, magnetic disturbance, rigid body motion

## Abstract

Normally, a three-dimensional orientation determination algorithm that is used in a magnetic and inertial measurement unit calculates the inclination (including both the pitch and roll) of rigid bodies by fusing the measurements of the gyroscope, as well as the measurements of both the accelerometer and the magnetometer. The measurements of the magnetometer can be helpful in improving the inclination estimation accuracy; however, once the measurements of the magnetometer are disturbed by ferromagnetic materials, the inclination estimation accuracy could be significantly decreased. Hence, a better approach should be followed in terms of not employing the measurements of the magnetometer for inclination determination. In order to achieve this goal, the component of the measurement of the magnetometer that is used for the improvement of the inclination estimation accuracy, along with the measurement of the accelerometer at each sampling time instant, is abandoned. Consequently, the remaining component of the measurement of the magnetometer, which is perpendicular to the measurement of the accelerometer, is used for the azimuth determination. After applying this process, the extended Kalman filter (EKF) is proposed for the inclination and azimuth estimations. Through experiments, the EKF is compared with three algorithms that were recently proposed with the same objective as this work, and the extracted outcomes show that the EKF approach clearly outperforms these three algorithms.

## 1. Introduction

This paper focuses on the decoupling problem faced by three-dimensional (3D) orientation determination algorithms (ODAs). ODAs are employed in magnetic and inertial measurement units (MIMUs) for determining their orientation when attached on moving rigid bodies, which is achieved by fusing the outputs of their magnetic, angular rate, and gravity (MARG) sensors. In fact, the basic principle of ODAs can be explained in a few words: the orientation is first predicted by relying on the measurement of the angular rate sensor and is then revised (or we can say “updated”) by introducing the measurements of the magnetic and gravity sensors. ODAs mainly differ by their different orientation revision strategies. The word “coupling” indicates that the measurement of the magnetic sensor participates in the calculation of the inclination angles, which causes the magnetic disturbance imposed on the magnetic sensor to degrade the estimation accuracy for the pitch and roll. Hence, the decoupling problem refers to understanding how to eliminate the negative effect of magnetic disturbance on the determination of inclination angles, including the pitch and roll. We present some novel modifications to the standard extended Kalman filter (EKF) for problem solving, showing that the EKF is still the most promising, powerful, and reliable ODA.

MIMUs are widely used in several applications where the determination of the inclination (including the pitch and roll) and azimuth (also called yaw or heading) of rigid bodies is required. Among the above-mentioned applications, the most conventional is the movement analysis of human body segments [1,2], followed by the inclination and azimuth sensing of small and medium-sized machines (usually autonomous equipment) flying through the air but near the ground [3], sailing under water [4,5], and driving on the ground [6]. Despite the fact that other methods, such as vision-based methods and GNSS-based methods, can also be applied for determining the inclination and azimuth, MIMUs have their unique superiorities, including their low cost, small size, light weight, and non-reliance on external artificial sources.

As mentioned, the key sensors within MIMUs are the tri-axis gyroscope (angular rate sensor), the tri-axis accelerometer (gravity sensor), and the tri-axis magnetometer (magnetic sensor), in addition to some other components such as the analog-to-digital converter, the temperature sensor used for sensor temperature drift compensation, and the micro-processor. A sensor fusion algorithm, such as a 3D ODA, is required in order to calculate the inclination and azimuth of the rigid body to which an MIMU is attached. Quaternions are usually selected for 3D orientation representations and are estimated in real time by applying an ODA, whereas the Euler angles are usually preferred when a visual presentation is needed. On top of that, if the latter takes place, then the quaternions should be converted to Euler angles.

ODAs can be further classified into two categories, i.e., Kalman filters and complementary filters. As far as Kalman filters are concerned, several modifications can be employed. More specifically, the linear Kalman filter (LKF) [7,8] and its extended version [9,10,11] have been reported. The latter incorporates the estimation of the gyroscope drift, while the former does not. Some other configurations have also been actively examined, including the EKF [12,13,14,15], the multiplicative extended Kalman filter (MEKF) [16,17,18,19,20], the unscented Kalman filter (UKF) [21,22], the cubature Kalman filter (CKF) [23], and the particle filter (PF) [24]. The structures of the EKFs proposed by [12,13,14,15] are exactly the same, except that the EKFs in [12] and [13] estimate any sensor bias, while the EKF in [15] estimates the sensor bias in addition to the orientation, and the EKF in [14] simplifies the Kalman gain matrix to decrease the computation time without reducing the orientation estimation accuracy. Different from the EKF, which realizes the linearization by using the Taylor expansion, the UKF and CKF realize the linearization by using unscented transformation and cubature transformation, respectively. The authors of the UKF claim that the approximations given by the UKF are accurate to the third order for Gaussian inputs for all nonlinearities, while they are accurate to at least the second order for non-Gaussian inputs [25]. The authors of the CKF claim that the CKF is more accurate and more principled in mathematical terms than the UKF [26]. The UKF, CKF, and PF are rarely used in MIMUs due to their high computational burden and relatively poor performance improvement compared to the EKF [27,28]. On the other hand, the MEKF is usually considered a first-order approximation to the EKF, and hence its accuracy is lower than that of the EKF [29]. The main difference among the various complementary filters is their diverse selection of single-frame inclination and azimuth determination algorithms. For instance, various algorithms have been investigated, including the Levenberg–Marquard algorithm [30], the factored quaternion algorithm [31], and the gradient descent algorithm [32,33,34,35]. It is interesting to note that complementary filters do not take into account the noise characteristics of the sensor measurements. As a result, their accuracy is normally lower than that of Kalman filters [36,37,38]. Moreover, the LKF exhibits a similar structure to complementary filters, except for the noise characteristics of the sensor measurements. In recent years, deep neural networks have also been introduced for inclination and azimuth determination, showing remarkable performances in some cases [39] due to their great success in pattern recognition.

It is well known that the measurements of the accelerometer are only useful for inclination determination, while the measurements of the magnetometer are mainly employed for azimuth determination. However, most ODAs take into account the measurements of both the accelerometer and magnetometer for inclination determination. Therefore, the inclination determination accuracy could be severely affected by the magnetic disturbance induced by ferromagnetic materials around MIMUs. In on-ground environments, particularly indoor environments, various hard and soft magnetic materials and excitation sources can be found, especially around the objects whose inclination and azimuth angles need to be determined, such as cars, security doors, mobile phones, metal tables and desks, and even whole buildings, which are obviously supported by ferroconcrete. MIMUs also contain magnetic materials and excitation sources. All these magnetic materials and excitation sources can distort the geomagnetic field. This means that large unmodeled magnetic disturbances occur more frequently than unmodeled acceleration disturbances; therefore, preventing magnetic disturbances from influencing the calculation of inclination angles will hopefully improve the estimation performance and hence is of practical importance. As a result, the goal of this work is to solve this problem. In the literature, various algorithms have been proposed in order to deal with this issue [40,41,42]. The basic idea of the extended complementary filter proposed in Ref. [40] is to correct the gyroscope drift by using the calculation results of two error terms, which are derived from the cross product of the measured and reference vectors for gravity and the geomagnetic field before integrating the gyroscope measurement to obtain the final orientation estimate. The four-parameter complementary filter presented in Ref. [41] is based on a new attitude representation, where 4D parameters (a 3D unit vector and one angle quantity) are used. The authors claim that the proposed representation has an intuitive interpretation and that the proposed filter has a separation property (magnetic disturbance does not affect the pitch and roll angle estimation). The fast complementary filter designed in Ref. [42] is a two-step filter. The pitch and roll are estimated in the first step, while the azimuth is estimated in the second step. Once it is recognized that a magnetic disturbance has emerged and degraded the accuracy of the magnetometer, the second step will be skipped so as to prevent the magnetic disturbance from affecting the computation of the pitch and roll. Obviously, a recognizer should be included in the designed filter; however, the recognizer cannot recognize all magnetic disturbances; hence, if it incorrectly recognizes a disturbance, the magnetic disturbance can still heavily affect the estimation accuracy of the pitch and roll.

From our experimental analysis, we found that the above-mentioned algorithms still present significant problems. Along these lines, we present various creative modifications to the EKF for problem solving. With the newly formulated measurement model and its error covariance matrix, the creatively modified EKF is capable of solving the decoupling problem, so that the negative effect of magnetic disturbance on the calculation of the inclination is eliminated. Experiments were carried out demonstrating that the modified EKF outperforms recently published algorithms. From our analysis, we draw the conclusion that the EKF is still the preferred algorithm when various practical problems are encountered.

The remainder of this paper is organized as follows: Section 2 elaborates on the details of the state transition model, as well as the measurement model that was used for the EKF. Section 3 provides the experimental comparison results between the EKF and three other algorithms that also aim to overcome the impact of magnetic disturbance on the inclination determination. Finally, Section 4 concludes this paper.

## 2. Analysis of the State Transition and Measurement Models That Were Used for the Extended Kalman Filter

### 2.1. Sensor Model

The outputs of the three-axis gyroscope, three-axis accelerometer, and three-axis magnetometer at the time instant $kT_{s}$ (where $T_{s}$ is the sampling period) are denoted as follows:(1)$$\left\{\begin{array}{l} \omega_{k}=\omega_{k}^{0}+\varepsilon_{k}^{\omega }\\ g_{k}=g_{k}^{0}+\eta_{k}^{g}+\varepsilon_{k}^{g}\\ m_{k}=m_{k}^{0}+\eta_{k}^{m}+\varepsilon_{k}^{m} \end{array}\right.$$
where *k* is a simple representation of “$kT_{s}$”, $\omega_{k}^{0}$ is the true angular velocity, and $\omega_{k}$ is its measurement provided by the gyroscope. Additionally, $g_{k}^{0}$ is the true unnormalized gravity represented in the body coordinate system, $g_{k}$ is its respective measurement given by the accelerometer, $m_{k}^{0}$ is the Earth’s true unnormalized magnetic field represented in the body coordinate system, and $m_{k}$ is its measurement value given by the magnetometer. Moreover, $\varepsilon_{k}^{\omega }$, $\varepsilon_{k}^{g}$, and $\varepsilon_{k}^{m}$ are assumed to be uncorrelated white Gaussian measurement noise, with a null mean and the covariance matrices $\Sigma_{\omega }=\sigma_{\omega }^{2}I$, $\Sigma_{g}=\sigma_{g}^{2}I$, and $\Sigma_{m}=\sigma_{m}^{2}I$ (where I is the identity matrix), respectively. Finally, $\eta_{k}^{g}$ and $\eta_{k}^{m}$ are the acceleration and magnetic disturbances, respectively.

### 2.2. State Transition Model

The continuous-time rigid-body angular motion is described by employing the following expression [43]:(2)$$\dot{q}=\frac{1}{2}\left[\begin{array}{cc} \left[\omega^{0}\times \right] & \omega^{0}\\ -{\left(\omega^{0}\right)}^{T} & 0 \end{array}\right]q$$
where $\omega^{0}={\left[\omega_{x}^{0},\omega_{y}^{0},\omega_{z}^{0}\right]}^{T}$, and the anti-symmetric matrix $\left[\omega^{0}\times \right]$ is constructed as follows:(3)$$[\omega^{0}\times ]=\left[\begin{array}{ccc} 0 & -\omega_{z}^{0} & \omega_{y}^{0}\\ \omega_{z}^{0} & 0 & -\omega_{x}^{0}\\ -\omega_{y}^{0} & \omega_{x}^{0} & 0 \end{array}\right]$$
where $\omega_{x}^{0}$, $\omega_{y}^{0}$, and $\omega_{z}^{0}$ are the true angular velocities along the X, Y, and Z axes of the body coordinate system, respectively. Furthermore, $q={\left[e^{T},q_{0}\right]}^{T}={\left[q_{1},q_{2},q_{3},q_{0}\right]}^{T}$ is the quaternion, whereas e and $q_{0}$ are its vector and scalar part, respectively.

By discretizing Equation (2), we obtain the following expression:(4)$$q_{k+1}=\exp\left(\frac{1}{2}\left[\begin{array}{cc} \left[\omega_{k}^{0}\times \right] & \omega_{k}^{0}\\ -{\left(\omega_{k}^{0}\right)}^{T} & 0 \end{array}\right]\right)q_{k}$$
where $q_{k}={\left[e_{k}^{T},q_{0,k}\right]}^{T}={\left[q_{1,k},q_{2,k},q_{3,k},q_{0,k}\right]}^{T}$ is defined as the quaternion at the time instant $kT_{s}$.

After the measurement noise is taken into consideration, Equation (4) is rewritten as follows:(5)$$q_{k+1}=\exp\left(\frac{1}{2}\left[\begin{array}{cc} \left[\omega_{k}\times \right] & \omega_{k}\\ -\omega_{k}^{T} & 0 \end{array}\right]\right)q_{k}-\frac{T_{s}}{2}\Xi_{k}\varepsilon_{k}^{\omega }$$
where $\Xi_{k}$ is expressed by the following equation:(6)$$\Xi_{k}=\left[\begin{array}{c} {}[e_{k}\times ]+q_{0\cdot k}I\\ -e_{k}^{T} \end{array}\right]$$

Equation (5) is actually the desired state transition model.

### 2.3. Measurement Model

The first measurement vector used here is $g_{k}$. The relation between $g_{k}$ and its reference unit vector $r_{g}={\left[0,0,-1\right]}^{T}$, which is represented in the global coordinate system (the local geographic coordinate system is selected, with its X, Y, and Z axes pointing north, west, and up, respectively), is expressed as follows:(7)$$g_{k}=g_{k}^{0}+\eta_{k}^{g}+\varepsilon_{k}^{g}=\left\|g_{k}^{0}\right\|C(q_{k}^{0})\left[\begin{array}{c} 0\\ 0\\ -1 \end{array}\right]+\eta_{k}^{g}+\varepsilon_{k}^{g}$$
where $\left\|g_{k}^{0}\right\|$ is the norm of $g_{k}^{0}$, $q_{k}^{0}$ is the true $q_{k}$, and the rotation matrix C(q) is defined by the following expression:(8)$$C(q)=\left[\begin{array}{ccc} q_{1}^{2}-q_{2}^{2}-q_{3}^{2}+q_{0}^{2} & 2(q_{1}q_{2}+q_{3}q_{0}) & 2(q_{1}q_{3}-q_{2}q_{0})\\ 2(q_{1}q_{2}-q_{3}q_{0}) & q_{2}^{2}-q_{1}^{2}-q_{3}^{2}+q_{0}^{2} & 2(q_{2}q_{3}+q_{1}q_{0})\\ 2(q_{1}q_{3}+q_{2}q_{0}) & 2(q_{2}q_{3}-q_{1}q_{0}) & q_{3}^{2}-q_{1}^{2}-q_{2}^{2}+q_{0}^{2} \end{array}\right]$$

The second measurement vector is normally $m_{k}$. However, in order to remove the influence of the magnetometer’s measurement on the inclination determination, its component along $g_{k}$ is not used. As a result, the second selected measurement vector is
(9)$$m_{k}^{\prime }=m_{k}-\frac{\left\|m_{k}\right\|}{\left\|g_{k}\right\|}g_{k}\cos(\theta_{gm})$$
where $\theta_{gm}$ is defined as the angle between $g_{k}$ and $m_{k}$.

The relation between $m_{k}^{\prime }$ and its reference unit vector $r_{m^{\prime }}={\left[1,0,0\right]}^{T}$, which is represented in the global coordinate system, is expressed as follows:(10)$$m_{k}^{\prime }={m_{k}^{\prime }}^{0}+\eta_{k}^{m^{\prime }}+\varepsilon_{k}^{m^{\prime }}=\left\|m_{k}^{0}\right\|\sin(\theta_{gm}^{0})C(q_{k}^{0})\left[\begin{array}{c} 1\\ 0\\ 0 \end{array}\right]+\eta_{k}^{m^{\prime }}+\varepsilon_{k}^{m^{\prime }}$$
where ${m_{k}^{\prime }}^{0}$, $\eta_{k}^{m^{\prime }}$, and $\varepsilon_{k}^{m^{\prime }}$ are, respectively, the components of $m_{k}^{0}$, $\eta_{k}^{m}$, and $\varepsilon_{k}^{m}$, and all are perpendicular to $g_{k}$. Furthermore, $\varepsilon_{k}^{m^{\prime }}$ is assumed to be white Gaussian measurement noise, with a null mean and the covariance matrix $\Sigma_{m^{\prime }}=\sigma_{m^{\prime }}^{2}I$; $\theta_{gm}^{0}$ is the true $\theta_{gm}$.

The proposed measurement model is constructed by stacking Equations (7) and (10) into one equation while ignoring the disturbance, as illustrated in the following expression:(11)$$\left[\begin{array}{c} \frac{g_{k}}{\left\|g_{k}\right\|}\\ \frac{{m^{\prime }}_{k}}{\left\|m_{k}\right\|\sin(\theta_{gm})} \end{array}\right]=\left[\begin{array}{cc} C(q_{k}) & 0_{3\times 3}\\ 0_{3\times 3} & C(q_{k}) \end{array}\right]\left[\begin{array}{c} r_{g}\\ r_{m^{\prime }} \end{array}\right]+\left[\begin{array}{c} \frac{\varepsilon_{k}^{g}}{\left\|g_{k}\right\|}\\ \frac{\varepsilon_{k}^{m^{\prime }}}{\left\|m_{k}\right\|\sin(\theta_{gm})} \end{array}\right]$$

In Equation (11), it is assumed that $C(q_{k}^{0})\approx C(q_{k})$, $\left\|g_{k}^{0}\right\|\approx \left\|g_{k}\right\|$, $\left\|m_{k}^{0}\right\|\approx \left\|m_{k}\right\|$, and $\theta_{gm}^{0}\approx \theta_{gm}$.

### 2.4. Calculation of the Error Covariance Matrices of the State Transition Model and the Measurement Model

The error covariance matrix of the state transition model given by Equation (5) is defined as follows:(12)$$Q_{k}={\left(\frac{T_{s}}{2}\right)}^{2}\sigma_{\omega }^{2}\Xi_{k}\Xi_{k}^{T}$$

By considering Equation (9), we can easily estimate that the error covariance matrix of the measurement model given by Equation (11) is formulated as follows:(13)$$R_{k}=\left[\begin{array}{cc} \frac{\sigma_{g}^{2}}{{\left\|g_{k}\right\|}^{2}}I & -\frac{\sigma_{g}^{2}}{{\left\|g_{k}\right\|}^{2}}\cot(\theta_{gm})I\\ -\frac{\sigma_{g}^{2}}{{\left\|g_{k}\right\|}^{2}}\cot(\theta_{gm})I & \frac{\sigma_{m^{\prime }}^{2}}{{\left\|m_{k}\right\|}^{2}{\left(\sin(\theta_{gm})\right)}^{2}}I \end{array}\right]$$
where $\sigma_{m^{\prime }}^{2}=\sigma_{m}^{2}+\frac{{\left\|m_{k}^{2}\right\|}^{2}}{{\left\|g_{k}^{2}\right\|}^{2}}\sigma_{g}^{2}{\left(\cos\left(\theta_{gm}\right)\right)}^{2}$. In order to derive Equation (13), it is assumed that $\theta_{gm}\approx \theta_{gm}^{0}$. However, the off-diagonal elements of $R_{k}$ that are provided by Equation (13) still render the estimation of the inclination susceptible to the measurements of the magnetometer; therefore, the matrix $R_{k}$ must be set as follows:(14)$$R_{k}=\left[\begin{array}{cc} \frac{\sigma_{g}^{2}}{{\left\|g_{k}\right\|}^{2}}I & 0_{3\times 3}\\ 0_{3\times 3} & \frac{\sigma_{m^{\prime }}^{2}}{{\left\|m_{k}\right\|}^{2}{\left(\sin(\theta_{gm})\right)}^{2}}I \end{array}\right]$$

By combining Equations (5), (11), (12) and (14), the EKF approach can then be used for the estimation of both the inclination and the azimuth [12,13,14,15].

## 3. Simulation, Experimental Results, and Discussion

Depending on the newly formulated measurement model and its error covariance matrix, shown in Equations (11) and (14), respectively, our creatively modified EKF now has the ability to solve the decoupling problem, that is, eliminating the negative effect of magnetic disturbance on the calculation of the inclination. According to the literature, the extended complementary filter (ECF) [40], the hybrid four-parameter complementary filter (FPCF) [41], and the fast complementary filter (FCF) [42] algorithms have the ability to prevent the magnetometer’s measurements from affecting the calculation of the inclination. In this section, the proposed EKF configuration will be compared with these three algorithms in order to assess the superiority of the former approach through the implementation of a thorough analysis.

### 3.1. Static Evaluation

In this test, the sensor outputs were given as follows:(15)$$\left\{\begin{array}{l} \omega_{k}=\varepsilon_{k}^{\omega }\\ g_{k}=\varepsilon_{k}^{g}\\ m_{k}=\varepsilon_{k}^{m} \end{array}\right.$$
where the covariance matrices $\Sigma_{\omega }$, $\Sigma_{g}$, and $\Sigma_{m}$ were set to the values of $0.006^{2}I$, $0.008^{2}I$, and $0.001^{2}I$, respectively. Moreover, the sampling time period $T_{s}$ was set to 1/100 s, $\left\|g_{k}^{0}\right\|=9{.8\;m/s}^{2}$, and $\left\|m_{k}^{0}\right\|=0.5\;\mathrm{Gs}(\mathrm{Gauss})$, whereas the true angle between $g_{k}^{0}$ and $m_{k}^{0}$ was $\theta_{gm}^{0}=32{}^{\circ}$, and the initial rotation matrix at time instant $kT_{s}=0$ was estimated using the following expression:(16)$$C(q_{0})=\left[\begin{array}{ccc} 0.9953 & -0.0772 & -0.0578\\ 0.0543 & 0.9441 & -0.3252\\ 0.0797 & 0.3206 & 0.9439 \end{array}\right]$$

The filter gains of both the ECF and FPCF are the only parameters that need to be optimally determined. Figure 1 depicts the distribution of the standard deviations of the estimation errors of the three Euler angles in conjunction with the changes in the filter gains $K_{\mathrm{norm}}$ and *a* of the ECF and FPCF, respectively. Moreover, Table 1 summarizes the minimal values of these standard deviations and their corresponding filter gains, which were obtained by implementing the ECF and FPCF algorithms. By setting the noise parameters of the EKF to $\sigma_{\omega }^{2}=0.006^{2}$, $\frac{\sigma_{g}^{2}}{{\left\|g_{k}\right\|}^{2}}=\frac{0.008^{2}}{9.8^{2}}$, and $\frac{\sigma_{m^{\prime }}^{2}}{{\left\|m_{k}\right\|}^{2}{\left(\sin(\theta_{gm})\right)}^{2}}=1.89\times \frac{0.001^{2}}{0.5^{2}}$, the standard deviations provided by the EKF were calculated, as summarized in Table 1. As can be ascertained, although the static performances of both the ECF and FPCF were promising, they did not approach the respective outcomes of the EKF.

### 3.2. Rotational Test by Incorporating Magnetic Disturbance

In this evaluation process, the sensor outputs were given as follows:(17)$$\left\{\begin{array}{l} \omega_{k}=\left\{\begin{array}{cc} \varepsilon_{k}^{\omega } & \;\;\;\;1\leq k<k_{1}\;\mathrm{or}\;k_{2}<k\leq k_{N}\\ \left[\begin{array}{c} \alpha_{x}^{\omega }\sin[\beta_{x}^{\omega }(k-k_{1})\pi T_{s}]\\ \alpha_{y}^{\omega }\sin[\beta_{y}^{\omega }(k-k_{1})\pi T_{s}]\\ \alpha_{z}^{\omega }\sin[\beta_{z}^{\omega }(k-k_{1})\pi T_{s}] \end{array}\right]+\varepsilon_{k}^{\omega } & \;\;\;\;k_{1}\leq k\leq k_{2} \end{array}\right.\\ g_{k}=\varepsilon_{k}^{g}\;\;\;\;\;\;\;\;\;\;\;\;\;\;\;\;\;\;\;\;\;\;1\leq k\leq k_{N}\\ m_{k}=\left\{\begin{array}{ll} \varepsilon_{k}^{m} & 1\leq k<k_{3}\;\mathrm{or}\;k_{4}<k\leq k_{N}\\ \mu_{k}^{m}+\varepsilon_{k}^{m}=\left[\begin{array}{c} \alpha_{x}^{m}\sin[\beta_{x}^{m}(k-k_{3})\pi T_{s}]\\ \alpha_{y}^{m}\sin[\beta_{y}^{m}(k-k_{3})\pi T_{s}]\\ \alpha_{z}^{m}\sin[\beta_{z}^{m}(k-k_{3})\pi T_{s}] \end{array}\right]+\varepsilon_{k}^{m} & k_{3}\leq k\leq k_{4} \end{array}\right. \end{array}\right.$$

In Equation (17), $k_{1}=500$, $k_{2}=2500$, $k_{3}=1000$, $k_{4}=3500$, $k_{N}=4000$, $\beta_{x}^{\omega }=3$, $\beta_{y}^{\omega }=4$, $\beta_{z}^{\omega }=5$, $\beta_{x}^{m}=1$, $\beta_{y}^{m}=2$, and $\beta_{z}^{m}=3$. The test was executed 2000 times, and in each run, $\alpha_{i}^{\omega }(i=x,y,z)$ and $\alpha_{i}^{m}(i=x,y,z)$ were randomly selected from the uniform distributions bounded by [0, 3] and [0, 0.5], respectively. The initial rotation matrix at a time instant of $kT_{s}=0$ was derived using Equation (16). Additionally, the filter gains of both the ECF and FPCF were set to $K_{norm}=2.4$ and a=6.6.

Figure 2 shows the extracted curves of the average absolute error of the inclination and azimuth estimates, which is defined as follows:(18)$$\xi_{u}(k)=\frac{1}{2000}\sum_{I=1}^{2000}\left|u_{est}(k)-u_{tru}(k)\right|\;\;1\leq k\leq k_{N}$$
where the letter “*u*” denotes the pitch, the roll, or the azimuth angle, $u_{est}$ signifies the estimate of the given Euler angle, and $u_{tru}$ represents the true value of this Euler angle.

From the calculated results presented in Figure 2, we can draw the conclusion that the inclination estimation accuracy achieved using both the ECF and FPCF approaches can still be affected by magnetic disturbance. Moreover, a deeper insight into this figure reveals that the magnetic disturbance effect may have a relationship with the rotational movement, since it can be clearly observed that magnetic-disturbance-induced accuracy degradation only occurred when a rotational movement took place (see the curves in the time interval 1000≤k≤2500). Additionally, in Figure 2, we can observe that the degradation of the inclination and azimuth estimation accuracy of the FPCF was also induced due to the impact of the rotational movements (see the curves in the time interval 500≤k≤1000).

In order to further explore the relationship between the inclination estimation accuracy, angular velocity, and magnetic disturbance, we employed Equation (17) in the following parts of this section by setting $\alpha_{x}^{\omega }=\alpha_{y}^{\omega }=\alpha_{z}^{\omega }$ and $\alpha_{x}^{m}=\alpha_{y}^{m}=\alpha_{z}^{m}$. The derived outcomes are shown in Figure 3.

As far as the ECF is concerned, it was found that the magnetic disturbance had a slight impact on its inclination estimation accuracy when the angular velocity was low, while the disturbance almost linearly decreased its inclination estimation accuracy when the angular velocity was high. This outcome implies that it may not be suitable to compensate the measurement error of the gyroscope by directly using the measurements of both the accelerometer and magnetometer. In other words, the normal practice, i.e., employing the measurements of both the accelerometer and the magnetometer in order to compensate for the prediction error of the inclination and azimuth calculated in the measurements of the gyroscope, may still be a better choice. As a result, different from the ECF approach, it is finally revealed that the rotational movement—not the magnetic disturbance—causes the degradation of the inclination estimation accuracy of the FPCF. Thus, as expected, magnetic disturbance does not affect the inclination estimation accuracy of the FPCF.

The authors of the FPCF method proposed a new inclination and azimuth representation, where 4D parameters (a 3D unit vector and one angle quantity) are used. However, the equation regarding the derivative of this new representation is nonlinear. Hence, in order to simply discretize the above-mentioned derivative equation, the authors used the first-order difference quotient to approximate the derivative equation, which explains why the inclination and azimuth estimations of the FPCF are susceptible to rotational movements.

### 3.3. Performance of FCF

The FCF is actually a two-step complementary filter, which fuses the measurements of the magnetometer in the second step for the update of the azimuth estimate. A comparative advantage of this algorithm is that, once a magnetic disturbance is recognized, the second step is not executed, and as a result, the influence of magnetic disturbance on the FCF can be suppressed.

Equation (19) is used in the FCF in order to determine whether a magnetic disturbance exists:(19)$$\left|\left|m_{k}\right|-\left|m_{k}^{0}\right|\right|\geq \alpha_{m}$$

The authors of the FCF did not comment on the exact value of $\alpha_{m}$ in their work; therefore, we set $\alpha_{m}=3\sigma_{m}$, which is based on the PauTa criterion. The filter gain of the FCF was set to $\gamma_{a}=\gamma_{m}=0.1$.

We still employed Equation (17) in this section by setting $\alpha_{x}^{\omega }=\alpha_{y}^{\omega }=\alpha_{z}^{\omega }=3$ and $\alpha_{x}^{m}=\alpha_{y}^{m}=\alpha_{z}^{m}=0.5$. Figure 4 illustrates the change curves of $\xi_{u}$ for the inclination and azimuth estimates. It is apparent that the impact of the magnetic disturbance on the FCF significantly depends on the judgment accuracy due to the existence of the magnetic disturbance. On top of that, if the movements are complicated, then it is inevitable that the magnetic disturbance could affect the inclination estimation accuracy due to the inaccurate judgment of its existence, as shown in Figure 4.

### 3.4. Performance Evaluation Using Real Sensor Measurements

In this test, an MIMU named MTi-G, which was created by Xsens (Henderson, NV, USA), was placed on a wooden table and kept motionless. After a while, a mobile phone, which was used in order to introduce magnetic disturbance, was moved around the MTi-G with the assistance of the tester’s hand. The standard deviations of the measurement errors of the gyroscope, accelerometer, and magnetometer were 0.0065 rad/s, $0{.01\;m/s}^{2}$, and 0.002 Gs, respectively. In addition, the sampling rate of the MTi-G was 10 Hz. Figure 5 depicts the inclination and azimuth estimation results provided by the EKF, ECF, and FPCF. The filter gains of the three algorithms were all optimally determined using the method explained in part A of this section. More specifically, they were set to the following values: $\sigma_{\omega }^{2}=0.0065^{2}$, $\frac{\sigma_{g}^{2}}{{\left\|g_{k}\right\|}^{2}}=\frac{0.01^{2}}{9.8^{2}}$, $\frac{\sigma_{m^{\prime }}^{2}}{{\left\|m_{k}\right\|}^{2}{\left(\sin(\theta_{gm})\right)}^{2}}=10\times \frac{0.002^{2}}{0.8^{2}}$, $K_{norm}=3.1$, and a=4.38.

Furthermore, the sensor measurements were down-sampled from 1 Hz to 9 Hz, and at each sampling rate, the filter gains of the EKF, ECF, and FPCF were optimally re-determined using the method explained in part A of this section. The standard deviations of the inclination estimation errors provided by the EKF, ECF, and FPCF at each sampling rate are highlighted in Figure 6. The standard deviation was computed as follows:(20)$$\sigma_{u}=\sqrt{\frac{1}{k_{N}-1}\sum_{k=1}^{k_{N}}{\left|u_{est}(k)-\mu \right|}^{2}}\;\;1\leq k\leq k_{N}$$
where $k_{N}=778$, and
(21)$$\mu =\sum_{k=1}^{k_{N}}u_{est}(k)\;\;1\leq k\leq k_{N}$$

The profiles of the inclination (pitch (upper left) and roll (upper right)) and azimuth (down middle) estimates provided by the EKF, ECF, and FPCF are given in Figure 5. As anticipated, it can be ascertained from Figure 5 that all three algorithms performed well, and only the azimuth estimation accuracy of these algorithms was affected by the magnetic disturbance. However, a deeper insight into the extracted data reveals that the azimuth estimation accuracy of the EKF seemed to be more immune to the magnetic disturbance compared with the other two algorithms.

As observed in Figure 6, after decreasing the sampling rate, the performances of both the ECF and FPCF deteriorated dramatically, while the performance of the EKF was only slightly affected. Hence, we can draw the conclusion that the EKF outperforms both the ECF and FPCF when the sampling rate is low.

Another MIMU, named MTi-300, was rotated almost periodically with the assistance of the tester’s hand. The rotation was performed around a computer case, which can distort the Earth’s magnetic field in the range of the rotation motion. An electromagnetic tracking system created by Polhemus (Colchester, CT, USA) was taken as the attitude reference. The standard deviations of the measurement errors of the gyroscope, accelerometer, and magnetometer were 0.002 rad/s, $0{.008\;m/s}^{2}$, and 0.001 Gs, respectively. In addition, the sampling rate of the MTi-300 was 100 Hz. Figure 7 depicts the inclination and azimuth estimation results provided by the EKF, ECF, and FPCF. The filter gains of the three algorithms were all optimally determined using the method explained in part A of this section. More specifically, they were set to the following values: $\sigma_{\omega }^{2}=0.002^{2}$, $\frac{\sigma_{g}^{2}}{{\left\|g_{k}\right\|}^{2}}=\frac{0.008^{2}}{9.8^{2}}$, $\frac{\sigma_{m^{\prime }}^{2}}{{\left\|m_{k}\right\|}^{2}{\left(\sin(\theta_{gm})\right)}^{2}}=26\times \frac{0.001^{2}}{0.8^{2}}$, $K_{norm}=1.6$, and a=7.1.

Since linear disturbance is introduced during the rotation motion, the filter gains should be dynamically regulated for linear disturbance suppression, as follows.
(22)$$\left\{\begin{array}{l} \sigma_{\omega }^{2}=\frac{1}{\mu }\sigma_{\omega }^{2}\\ \frac{\sigma_{g}^{2}}{{\left\|g_{k}\right\|}^{2}}=\mu \frac{\sigma_{g}^{2}}{{\left\|g_{k}\right\|}^{2}}\\ \frac{\sigma_{m^{\prime }}^{2}}{{\left\|m_{k}\right\|}^{2}{\left(\sin(\theta_{gm})\right)}^{2}}=\mu \frac{\sigma_{m^{\prime }}^{2}}{{\left\|m_{k}\right\|}^{2}{\left(\sin(\theta_{gm})\right)}^{2}}\\ K_{norm}=\mu K_{norm}\\ a=\frac{1}{\mu }a \end{array}\right.$$

The scalar μ is determined according to the following:(23)$$\mu =\left\{\begin{array}{ll} 1, & \left|\left|\sum_{i=k-N+1}^{k}g_{i}\right|-\left|g_{k}^{0}\right|\right|\leq 3\sqrt{3}\sigma_{g}\\ 10^{100}, & \left|\left|\sum_{i=k-N+1}^{k}g_{i}\right|-\left|g_{k}^{0}\right|\right|>3\sqrt{3}\sigma_{g} \end{array}\right.$$
where N=20.

Obviously, the negative effect caused by the linear acceleration disturbance is suppressed, but the negative effect caused by the magnetic disturbance is completely remains. The profiles of the inclination (pitch (upper left) and roll (upper right)) and azimuth (down middle) estimates provided by the EKF, ECF, and FPCF are given in Figure 7.

As anticipated, it can be ascertained from Figure 7 that the EKF behaves basically the same as the ECF, except that the dynamic response of the EKF is a bit faster than that of the ECF. However, the azimuth estimation accuracy of the FPCF is significantly decreased. The FPCF heavily relies on the gyroscope measurements for attitude estimation when Equation (22) is employed. As mentioned above, the authors of the FPCF used the first-order difference quotient to approximate the derivative equation, which may explain the poor azimuth estimation performance. In addition, the magnetic disturbance further deteriorates the azimuth estimation performance. It is noted that the orientation (including the pitch, roll, and azimuth) estimation errors shown in Figure 7 are mostly caused by the gyroscope measurement error and the comparatively low sampling rate, and the azimuth estimation error is partly caused by the magnetic disturbance. Due to the similarity between the pitch and roll estimation accuracy, it is deduced that the magnetic disturbance does not negatively affect the pitch and roll calculation of the EKF, ECF, and FPCF.

## 4. Conclusions

In this work, we have demonstrated that the EKF is an important alternative solution in applications where the influence of magnetic disturbance on the inclination estimation is not acceptable. Despite the fact that three algorithms, i.e., ECF, FPCF, and FCF, have recently been proposed, there are still significant issues that should be addressed. The ECF is susceptible to magnetic disturbance when a large angular movement takes place. Moreover, the FPCF cannot perform well when only a large angular movement exists, regardless of whether a magnetic disturbance exists or not. Additionally, both the ECF and FPCF exhibit poor performance when the sampling rate is decreased. The impact of magnetic disturbance on the FCF strongly depends on the identification accuracy for the existence of magnetic disturbance. With some modifications, the EKF not only achieves the goal of removing the influence of the magnetometer’s measurements on the inclination determination, but also overcomes the above-mentioned issues faced by the ECF, FPCF, and FCF algorithms. Hence, it is concluded that the EKF can be selected for the related applications described in our work, and is still the preferred algorithm for determining the attitude of rigid bodies.

## Figures and Tables

**Figure 1 sensors-23-09756-f001:**
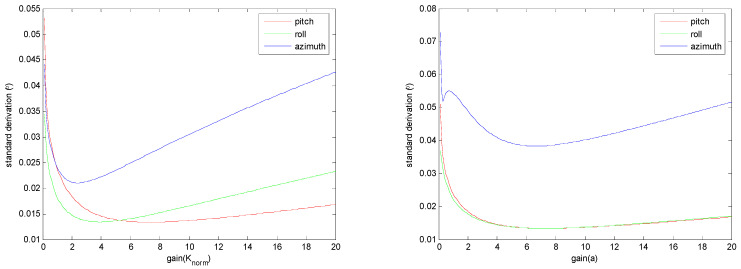
Distribution of the standard deviations of the estimation errors of the three Euler angles as a function of the modifications of the filter gain $K_{\mathrm{norm}}$ of the ECF (**left**) and the filter gain *a* of the FPCF (**right**).

**Figure 2 sensors-23-09756-f002:**
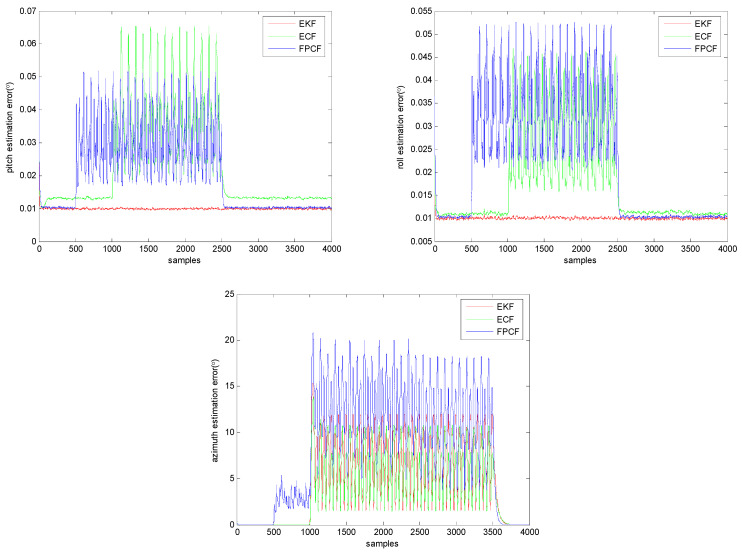
Variation curves of the average absolute errors of the inclination (pitch (**upper left**) and roll (**upper right**)) and azimuth (**down middle**) estimates calculated using the EKF, ECF, and FPCF.

**Figure 3 sensors-23-09756-f003:**
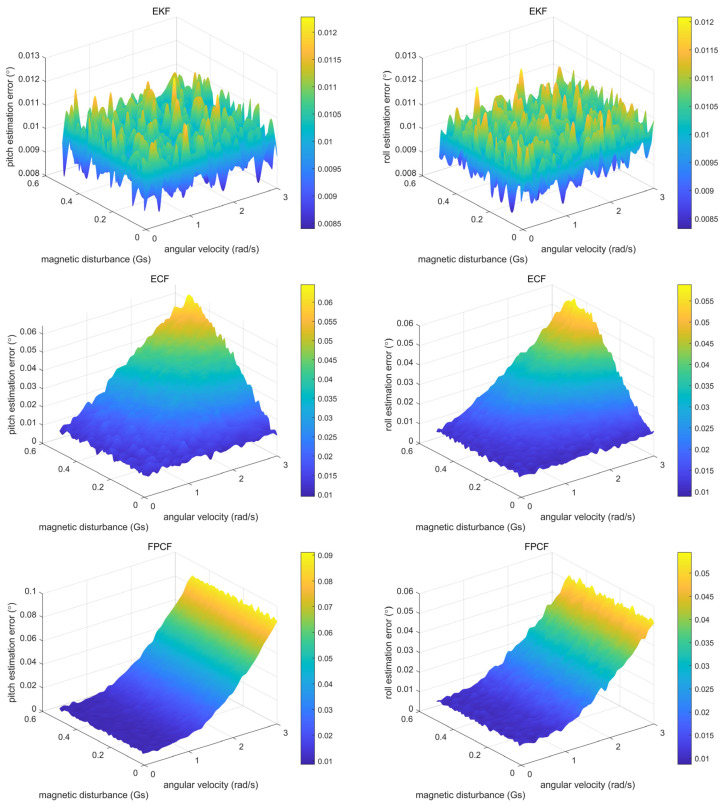
Curved surfaces describing the changes in the pitch and roll estimation errors of the proposed EKF, ECF, and FPCF, along with the changes in the magnitude of the angular velocity, and the magnetic disturbance.

**Figure 4 sensors-23-09756-f004:**
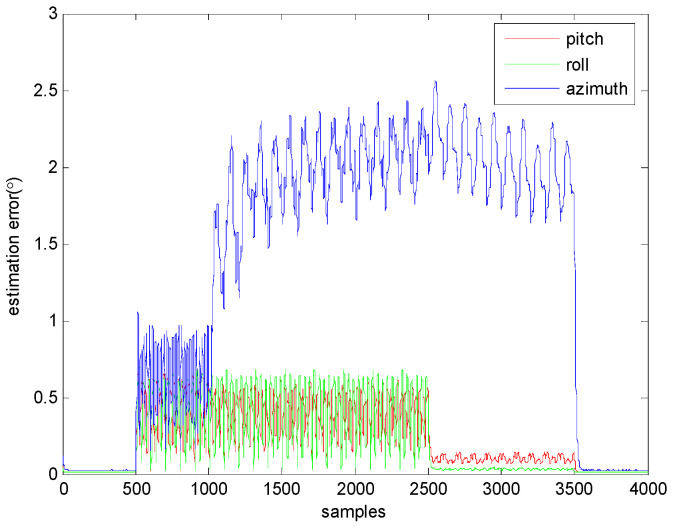
Time-related change curves of the inclination and azimuth estimation errors of the FCF.

**Figure 5 sensors-23-09756-f005:**
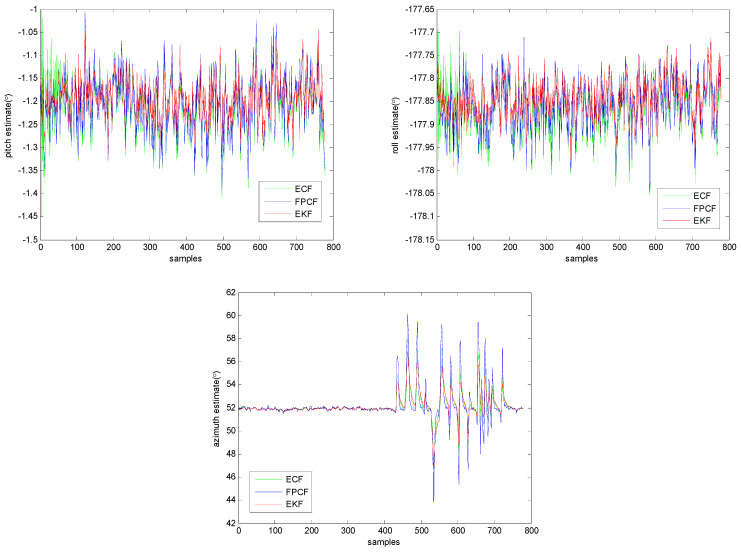
Profiles of the inclination (pitch (**upper left**) and roll (**upper right**)) and azimuth (**down middle**) estimates provided by the EKF, ECF, and FPCF when MTi−G is employed for a real test.

**Figure 6 sensors-23-09756-f006:**
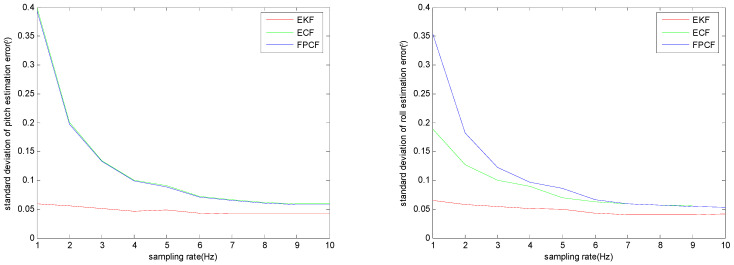
Distribution of the standard deviations of the inclination (pitch (**left**) and roll (**right**)) estimation errors provided by the EKF, ECF, and FPCF.

**Figure 7 sensors-23-09756-f007:**
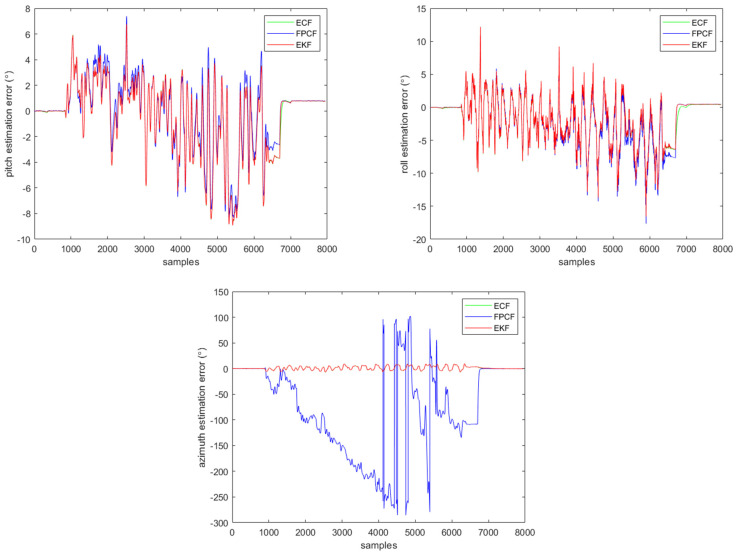
Profiles of the inclination (pitch (**upper left**) and roll (**upper right**)) and azimuth (**down middle**) estimates provided by the EKF, ECF, and FPCF when MTi−300 is employed for a real test.

**Table 1 sensors-23-09756-t001:** Minimal values of standard deviations of the estimation errors of the three Euler angles that were estimated using the EKF, ECF, and FPCF approaches.

	EKF	ECF	FPCF
pitch	0.0129	0.0134 (*K*_norm_ = 7.3)	0.0134 (*a* = 7.4)
roll	0.0130	0.0134 (*K*_norm_ = 3.9)	0.0134 (*a* = 7.2)
azimuth	0.0213	0.0210 (*K*_norm_ = 2.4)	0.0383 (*a* = 6.6)

## Data Availability

The data generated and/or analyzed in the current study are not publicly available for legal/ethical reasons but are available from the corresponding author upon reasonable request.

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
