# Peer review of "A Customized Extended Kalman Filter for Removing the Impact of the Magnetometer’s Measurements on Inclination Determination"

_sensors, 2023, doi:10.3390/s23249756_

Round 1

Reviewer 1 Report

Comments and Suggestions for Authors

It is unclear that the measurement scenarios will be. Why it is necessary to consider about the magnetic disturbances of the environment. 

A thorough literature should have been done considering different measurement applications such as using IMU for human motion monitoring and robotic measurements, navigation etc. Below is a list of references:

10.3390/rs15020533

The experiments were not designed properly. A controlled turntable can be used for better estimation of the changes in gyro. Also the gold standard measurement is missing from this study, a camera based system can be used to provide movement reference. 

More description and a system set-up should be presented. A MTi-G is used in the experiment. Have you compared the measurement results from the sensor with the results by your proposed algorithm.

Reviewer 2 Report

Comments and Suggestions for Authors

This paper compared the performance of EKF, ECF, FPCF, FCF used in 3D-ODA. It concluded that EKF was the preferred algorithm.

(1)  What is the contribution of the paper, that other researchers can referenced?

(2)  ECF, FPCF, FCF were seldom described in Introduction.

(3)  A sample frequency of 100Hz was selected in 3.1 static evaluation. Why the frequency range was 1-10Hz in Fig.6?

(4)  In line 264, Is “Fig.5” correct?

(5)  In section 3.4, how the standard deviations of the measurement error were determined?

Comments on the Quality of English Language

The English language can be improved .

Reviewer 3 Report

Comments and Suggestions for Authors

Ms. ID sensors-2651032-peer-review-v1, submitted to MDPI “Sensors”, Title: “A Customized Extended Kalman Filter for Removing the Impact of the Magnetometer’s Measurements on Inclination Determination, By: Yang Chen and Hailong Rong

Comments and Suggestions

In this manuscript, authors deal with the sensor algorithms attached in rigid bodies aiming at the real-time tracking of their motion. These sensors correspond to the tri-axis gyroscope, the tri-axis accelerometer, and the tri-axis magnetometer. Measurements of accelerometer provide the inclination determination, while those of magnetometer provide the azimuth-determination. Authors, in order to improve the accuracy of the employed algorithms, propose a new version of the known “Extended Kalman Filter (EKF) algorithm” which reduces the impact of the magnetic disturbance on the inclination determination. The performance of the new version is compared with three other algorithms existed in the literature.

In the manuscript’s subject fall quite interesting applications related to the motion of robotics, underwater and aerospace vehicles, etc. (also, the human body can be viewed as an articulated rigid body, e.g. skull’s motion is observed via equipment of inertial sensor systems). The manuscript, however, though well organized, has not been very carefully written and needs improvements at least in the points described below.

1) The phrase “Rigid body motion” is a key concept for the paper’s topic and should necessarily appear (in addition to Section 2) also (at least): (i) in the paper’s abstract, (ii) in the paper’s “Keywords”, and (iii) in the Conclusions Section.

2) At the beginning of the Introduction, a new paragraph must be inserted for introducing the reader into the paper’s topic and provide general information on the sensor algorithms attached on the rigid bodies. For example, some of the existing concepts in the 3rd paragraph and elsewhere in the Introduction must be discussed in the first paragraph.

3) The proposed by the authors “Extended Kalman Filter (EKF)” for the inclination and azimuth determination, as it is mentioned in the manuscript, had already been addressed by the authors of Refs. [11-14]. Please, in the Introduction and in the first paragraph of Sect. 3, clarify and stress the novelty and characteristics introduced in the EKF by your present work (e.g. previous improvements of EKF are distinguished among each other by their different names).

In summary, the paper is interesting but authors must make the improvements suggested above (see also the minor points below) before I recommend its publication in the J. “Sensors” of MDPI.

Minor Points

-- Authors, please, improve throughout (i) the punctuation and (ii) the use of “Tab”, specifically just after equations. The text looks as having a “very large number” of paragraphs (in LATEX one can avoid it by using the command “\noindent” before each paragraph).

-- Please, for the sake of completeness for young researchers, put a reference just before Eq. (2).

-- In P-2 L-90, after Eq. (1), please give details about the values of index k (at least add a comment there).

-- In P-2 L-90, please, correct: “.. mag netometer …’’ ==> “... magnetometer …’’

-- In P-3 L-97, please, correct the ...index g --→ m: “..

-- In P-3, L106, please, put “whitespace” in the cases needed.. Please, check the entire manuscript since a plethora of such cases exists.

– In P-5, L-158, please, correct: Do you mean, 11-14 ==> [11-14]?

-- In the “Conclusions” Section, the first two sentences do not add something valuable. In addition, we very rarely use citations in the “Conclusions” Sect. I suggest either to delete them completely or to move them (only as one parenthetical and shorter sentence) at the end of the previous Section.

-- P-13, L-309, please, change: “As has been demonstrated in this work, EKF can still be an important alternative solution in applications where the influence of magnetic disturbance on the inclination estimation is not acceptable”, ==> “In this work, we have demonstrated that, EKF is an important alternative solution in applications where the influence of magnetic disturbance on the inclination estimation is not acceptable.”

-- In Eq. (18), please, check if the symbol AAEu is consistent with the Journal’s policy (too many capital letters!)

– Please, improve Figure captions of Fig. 3 and Fig. 4.

-- For reader’s convenience (also in order to shorten paper’s size), I suggest: the two sub-Figures in Figs. 1 and 6 as well as the first two sub-Figures of Figs. 2 and 5 to be inserted side-by-side, and not the one below the other.

– Everywhere in the text, please, follow Journal’s policy, e.g. “Eqn. (4)” ==> “Eq. (4)” …

– Kindly please, read and correct very carefully the entire paper, at least two times.

Comments on the Quality of English Language

The paper needs extensive editing of English language.

In addition, Journal's policy is not followed well by

authors (see my report).

Round 2

Reviewer 1 Report

Comments and Suggestions for Authors

I am happy with the revision.

Reviewer 3 Report

Comments and Suggestions for Authors

Ms. ID sensors-2651032-Revised_V2, submitted to MDPI “Sensors”, Title: “A Customized Extended Kalman Filter for Removing the Impact of the Magnetometer’s Measurements on Inclination Determination”, By: Yang Chen and Hailong Rong

Comments and Final Recommendation

The authors of the above referenced manuscript, have addressed successfully and to a great extent the concerns raised in the Review Report of the initially submitted version. The revised version of the paper appears clearly improved mainly in the following points.

The authors added some new necessary paragraphs in the Introduction for improving the state of the art of the paper’s subject and for clarifying the novelties of the manuscript. Moreover, authors added some (mostly smaller) paragraphs throughout the text and included two additional references in the manuscript’s reference list. As a result of these modifications the paper now appears clearly more complete and greater in size.

Further, authors proceeded with the better organization of the figures which not only saves space but it is beneficial for the reader since it facilitates comparisons among the various sub-figures. Also, now the figure captions have been improved.

Finally, authors made official English language MDPI editing, which means that the manuscript has been checked for correct use of grammar and common technical terms. This makes the task of Journal proofreaders easier.

Based on the above improvements and modifications, I recommend “acceptance as it is” of the paper for publication in the Journal “Sensors” of MDPI as a regular article.